# Full Stack Optimization of Transformer Inference

Sehoon Kim[*1], Coleman Hooper[*1], Thanakul Wattanawong[1], Minwoo Kang[1], Ruohan Yan[1], Hasan Genc[1]
Grace Dinh[1], Qijing Huang[2], Kurt Keutzer[1], Michael W. Mahoney[134], Yakun Sophia Shao[1], Amir Gholami[13]

[1]University of California, Berkeley   [2]NVIDIA   [3]ICSI   [4]LBNL

*Abstract*—**Recent advances in state-of-the-art neural network architecture design have been moving toward Transformer models. These models achieve superior accuracy across a wide range of applications in computer vision, natural language processing, and speech recognition. This trend has been consistent over the past several years since Transformer models were originally introduced. However, the amount of compute and bandwidth required for inference of recent Transformer models is growing at a significant rate, and this has made their deployment in latency-sensitive applications challenging. As such, there has been an increased focus on making Transformer models more efficient, with methods that range from changing the architecture design, all the way to developing dedicated domain-specific accelerators.**

**In this work, we pursue a full-stack approach to optimizing Transformer inference. We analyze the implications of the Transformer architecture on hardware, including the impact of nonlinear operations such as Layer Normalization, Softmax, and GELU, as well as linear operations, and we use this analysis to optimize a fixed Transformer architecture. We assess the challenges with finding the right mapping and scheduling of operations for Transformer models, and pursue neural architecture search to further optimize the Transformer network. We find that a full-stack co-design approach with the aforementioned methods can result in up to 88.7× end-to-end speedup with minimal performance degradation for Transformer inference. More details can be found in our full paper [27], which includes (1) a comprehensive analysis of Transformer workloads, (2) an extensive survey of the current hardware and software solutions on efficient Transformer inference, and (3) case studies to quantify the advantages of co-design and co-optimization techniques across the stack on full-stack Transformer inference.**

## I. INTRODUCTION

Deep learning models have scaled up to billions of parameters and billions of multiply-accumulate operations during both training and inference. As a result, there has been a growing interest in computing these models efficiently and in deploying these compute and memory-intensive workloads on resource-constrained edge devices. These edge devices have tight energy and memory constraints, and the corresponding applications that leverage deep learning models also often have real-time latency constraints.

The demand for fast and efficient computation, coupled with the characteristics of deep learning workloads that involve a small set of distinct operations with substantial data reuse, have led to the use of hardware accelerators. A multitude of enterprise deep learning accelerators, such as [1], [3], [17], [23], [25], [28]–[30], [37], [44], [46], have been developed and integrated into commodity hardware by industry in the past decade. This parallels many research accelerators developed in academia [7]–[10], [16], [18]–[20], [36]. Together with hardware accelerator development, the software frameworks [2], [5], [24], [34] and compilers [6], [32], [42] for deploying various deep learning algorithms have also enhanced and matured. These tools enable the execution of deep learning algorithms on accelerators, and they perform mapping optimizations to improve the performance and efficiency of the full deep learning pipeline. Nonetheless, the fast-evolving deep learning algorithms still keep introducing new demands for hardware and software support, as well as their co-optimization, to satisfy various deployment constraints.

The recent rise in popularity of Transformers and large language models [4], [12], [14], [15], [21], [38]–[41], [43], [45] for solving various natural language processing (NLP) tasks presents a brand new set of challenges in the design of accelerators as well as frameworks. There has been an increased focus on making Transformer inference more efficient, especially due to their growing size and run-time complexity. However, there is still a lack of understanding regarding the workload characteristics of Transformer architectures, and thus of the design principles necessary for effectively running these models, when compared to the more well-known convolutional neural network (CNN) architectures. For instance, compared to the conventional CNN-focused design, Transformers are mostly composed of matrix multiplications (matmuls) together with memory-intensive nonlinear operations. In addition, the computational graph and dataflow of Transformer models are more complex than that of CNNs, with more types of operation nodes, as well as more dataflow splits and concatenations. All these challenges require us to undertake a comprehensive analysis of the current hardware and software solutions as well as the various design trade-offs for Transformer inference.

Our analysis yielded several key findings:
- We adapt Gemmini [19], which was originally designed for CNN workloads, for Transformer inference. Without modifications, the primary bottleneck for running Transformers on CNN accelerators is the time spent on floating-point non-linear operations. However, by adapting Gemmini to support an integer-only BERT variant [26], and tuning the memory configuration, we improve performance by 39.6×.
- Fusing BatchNorm with the neighboring convolution in CNNs is straightforward. However, the benefits of fusing operations in the Transformer architecture with the preceding matmuls depends on the particular operation as it can impose constraints on the mapping, leading to runtime

*Equal contribution. sehoonkim@berkeley.edu, chooper@berkeley.edu

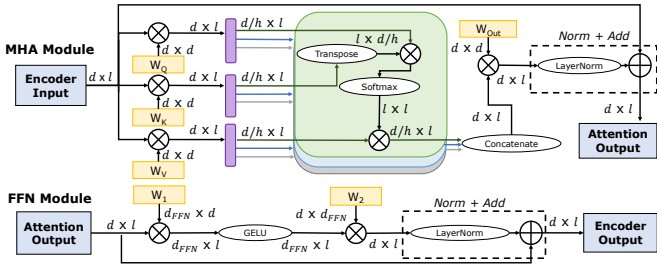

**Fig. 1:** *Map of the computations performed in (Top) the multi-head attention (MHA) module and (Bottom) the feed-forward network (FFN) module in the Transformer encoder block*

costs that outweigh the gains from operator fusion.
- We apply automated neural architecture search (NAS) to search for efficient and high-performance Transformer architectures on Gemmini-driven hardware. NAS finds an architecture that improves EDP by $10.6\times$ with minimal degradation on target benchmark. Combined with the hardware improvement, we achieve $88.7\times$ end-to-end speedup.

## II. HARDWARE ARCHITECTURE OPTIMIZATION

We first illustrate how architects familiar with mainstream accelerators for convolutional, vision-based workloads can design state-of-the-art Transformer accelerators. We start with a fairly typical CNN accelerator generated by the Gemmini [19] accelerator-generator, optimized primarily for ResNet50-like workloads, and we discuss changes we made to this accelerator and its software stack to efficiently support Transformer workloads such as BERT. Throughout this section, we use BERT-Base as a workload. For more details, please refer to Section 3 of our full paper [27].

*1) Baseline Accelerator:* We first generate a fairly typical CNN accelerator with a $16\times16$ systolic array and the weight-stationary dataflow using the Gemmini accelerator-generator. The 8-bit integer weights and inputs are stored in a 256 kB local scratchpad memory, and the 32-bit partial sums are stored in a dual-ported 64 kB accumulator SRAM which performs matrix additions. When DNN layers are too large to fit into the local scratchpad, they fall back onto an external L2 cache and DRAM which are shared with CPUs and other accelerators on the system-on-chip (SoC). A host CPU tiles such layers to compute the full outputs. The baseline accelerator produced by Gemmini incorporates peripheral circuitry that enables the execution of ReLU and max-pool operations, alongside integer-float multipliers that facilitate the scaling of 32-bit partial sums into 8-bit inputs for the subsequent layer. Native support for these operations is important, as it eliminates the necessity of offloading such operations to the host CPUs, thereby circumventing the costly transfers of activations between DRAM or outer caches and the local scratchpad.

Finally, note that this baseline CNN accelerator does not include any Transformer-specific features. In particular, there is no support for nonlinear normalization operations such as GELU, Softmax, or LayerNorm. Therefore, although it

achieves real-time or near-real-time performance on end-to-end CNN workloads, the performance on Transformer workloads such as BERT is severely limited [19] as will be discussed in more detail.

*2) Performance Bottlenecks:* Our observation has revealed that the baseline CNN accelerator, when deployed for Transformer inference, exhibits $< 1\%$ utilization of its functional units. Although individual matmuls exhibit 74% utilization, the performance is severely impeded by the nonlinear operations that need to be executed on the host CPU as they are not natively supported by the accelerator. This is further exacerbated by the fact that the nonlinear operations necessitate the use of floating-point arithmetic. Not only it is less energy and latency efficient than their integer counterparts [22], it also entails dequantization and re-quantization of the activations. These overheads account for 96% of the overall execution time (Fig. 2). Given that the majority of FLOPs in Transformer inference are matmuls, the time spent on the nonlinear operations in the baseline accelerator is far from the theoretical optimal, unless further optimizations are implemented.

In contrast to the convolutions in CNNs, which exhibit high arithmetic intensity, Transformers mostly comprise matmuls, often with small and/or rectangular matrices, which translate to lower arithmetic intensities and different optimal tiling strategies. This indicates that the memory hierarchy and memory bandwidth of our baseline CNN accelerator need to be recalibrated for more efficient Transformer inference.

*3) Memory Configuration Re-adjustment:* We have observed that the performance of BERT matmul operations can be significantly improved by adjusting the sizes of the input/weight scratchpad and the partial sum accumulator. Specifically, we have found that larger accumulators with higher output-reuse are more suitable for several matmuls in Transformers, such as the query $\times$ key matmuls, which have $l \times l$ output activation matrices which can be much larger than the $l \times d/h$ input matrices for $l$, $d$, and $h$ sequence length, hidden dimension, and number of heads, respectively. Based on this observation, we have modified the CNN-optimized memory configuration of our baseline accelerator by reducing the size of the scratchpad from 256 kB to 64 kB, and increasing the size of the accumulator from 64 kB to 256 kB. Importantly, these changes do not result in an increase in the total SRAM capacity or the total area; however, they result in a substantial 36% reduction in total matmul latency.

*4) Hardware-Software Co-Design:* To alleviate the overhead incurred by runtime quantization and dequantization, as well as the offloading of nonlinear operations to the CPU, we have transitioned our baseline Transformer workload from a naive BERT implementation, where only matmuls are quantized, to an integer-only BERT variant known as I-BERT [26]. I-BERT substitutes floating-point nonlinear operations with integer polynomial approximations, which can be implemented faster and more efficiently in specialized accelerators. To incorporate I-BERT, we add new integer implementations of I-BERT's GELU, LayerNorm, and Softmax variants to our

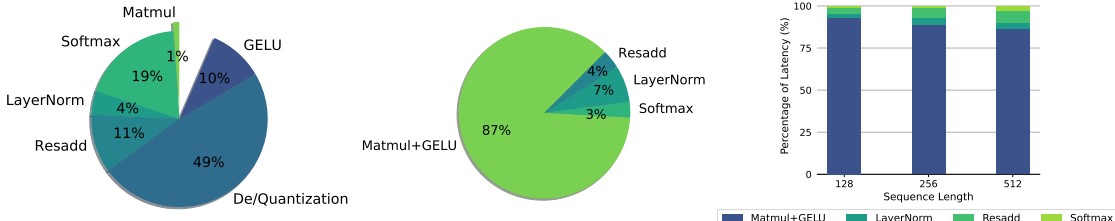

**Fig. 2:** *The time breakdown of a BERT inference with a sequence-length of 512, when running on (Left) the baseline CNN accelerator, and (Middle) the accelerator with I-BERT's hardware/software features incorporated. (Right) The time breakdown with different sequence lengths after the change. For all sequence lengths, the total execution time is dominated by matmuls.*

baseline CNN accelerator. The 32-bit matmul results residing in the accumulator are fed into a newly added "normalization unit" which computes reduction operations (e.g. sum, sum-of-square, max, etc.) which are used by LayerNorm and Softmax. Multiple passes of accumulator reads are required to compute all the reductions in these operations. Subsequentially, the matmul results in the accumulator undergo a final read operation to be fed into a set of 16 activation units, which compute I-BERT's non-linear variants in parallel.

With these new features, overall end-to-end BERT inference performance improved by 39.6× over the baseline accelerator's initial performance. As Fig. 2 illustrates, the computational bottleneck once again became the matmuls rather than normalization or activation functions. Quantization and dequantization no longer become necessary and GELU can be trivially fused with the preceding matmuls, so that they become one pipelined operation. When synthesized with the ASAP7 PDK [13], the new hardware units increased the total area consumption of the accelerator by only 14%, and the GELU, LayerNorm, and Softmax operations increased the power consumption of a BERT inference by only 9.3%.

## III. Scheduling Optimization

In Sec. II, we have demonstrated that the nonlinear operations in Transformers introduce challenges to efficient accelerator design. We further find that these operations present non-trivial challenges to the scheduling problem as well. In this section, we provide a brief overview of those challenges. For more details, please refer to Section 5 of our full paper [27].

Generally in DNN scheduling, it is an enticing strategy to fuse relatively high-arithmetic-intensity matmuls with the following low-arithmetic-intensity normalization operations. For example, execution schedulers for CNN-type accelerators often fuse convolutions with ReLU or max-pool operations. This strategy is especially applicable in the case of quantized workloads, where partial sums awaiting normalization are often of higher bitwidth than the final normalized outputs.

Similarly, for Transformer encoders, we could overlap the execution of normalization operations (LayerNorm and Softmax) with their preceding matmuls. However, this strategy may require hardware/software changes. First in the case of DNN accelerators like Gemmini, additional hardware support for directly accessing partial sums by normalization operation units may be required. Second, appropriate constraints on the

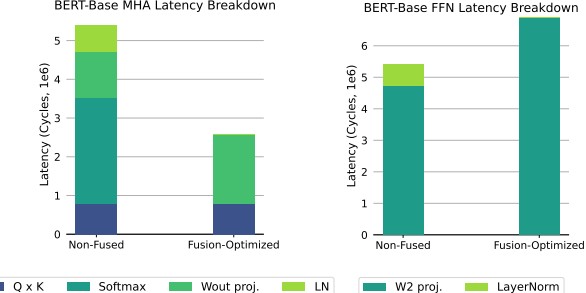

**Fig. 3:** *(Left) Impact of fusion-optimized scheduling for MHA execution. Hiding the Softmax latency via fusion-optimized scheduling improves overall MHA latency by 78%, but overlapping $W_{out}$ projection with LayerNorm can hurt total latency. (Right) Impact of fusion-optimized scheduling for FFN matmul that enables latency hiding of the LayerNorm operation. We observe that fusion-optimized scheduling hurts total latency by 27%. In both cases, we assume an input sequence length of 512 and accumulator size of 256kB.*

matmul execution schedule are necessary. In particular, the tiling factor size of either output dimension of the matmul must be maximized, so that rows/columns are immediately ready and stored at the Gemmini accumulator scratchpad for computing the mean and standard deviation. We refer to this alternate scheduling approach as *fusion-optimized scheduling*.

In Fig. 3, we take a deeper look into the performance implications of fusion-optimized scheduling for the BERT-Base encoder. We model the total latency of each adjacent pair of matmul and LayerNorm/Softmax operations via Timeloop [33] with the target hardware being the I-BERT modified Gemmini described in Sec. II. Opportunities for overlapping computations include: (1) the MHA query × key matmul and following Softmax; (2) MHA $W_{out}$ projection and following LayerNorm; and (3) FFN $W_2$ projection and following LayerNorm. The two scheduling strategies we compare are: (1) fusion-optimized scheduling and (2) Gemmini's default heuristic-based scheduler, which greedily maximizes loop tile factors at the local SRAM level for each of the three matmul dimensions. We refer to the second, default scheduling approach as *non-fused* scheduling.

The left plot of Fig. 3 showcases the promises of matmul and non-linear operator fusion within the MHA. With Gemmini on-chip scratchpad and accumulator SRAM sizes of

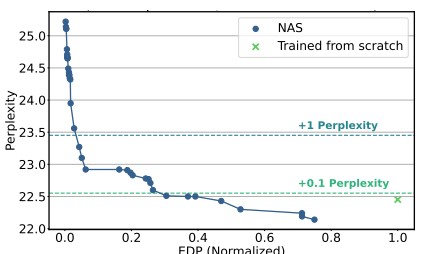 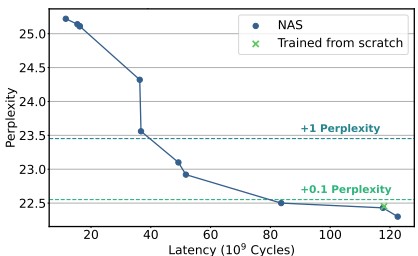 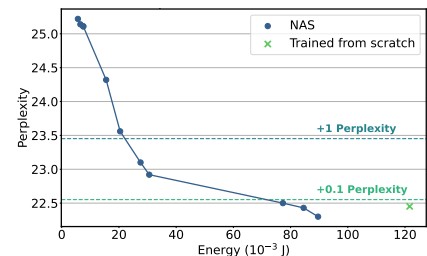

**Fig. 4:** *(Left) EDP-perplexity, (Middle) Latency-perplexity, and (Right) Energy-perplexity plots of the Transformer architectures found via evolutionary search on our Gemmini hardware configuration. Lower perplexity indicates better performance of the trained models. For better comparison, we additionally plot lines to illustrate +0.1 and +1 point perplexity degradation.*

256KB, we observe that it is advantageous to fuse query × key matmuls with Softmax for each attention head and thereby hide the relatively high latency of executing the Softmax operation. Assuming an input sequence length of 512, the Softmax latency is significant compared to the matmul, taking up around 78% of the total cycles and contributes greatly to the total latency.

On the other hand, the right plot of Fig. 3 shows the results on matmul and LayerNorm overlapping in the FFN $W_2$ projection. Here, we observe that fusion-optimized scheduling worsens total latency by 27%. When scheduling the FFN, we find that at the BERT-Base scale, it is consistently favorable to overlap the MHA query × key with the ensuing Softmax but consistently disadvantageous to chain the FFN $W_2$ projection matmul with LayerNorm. This is in contrast with previous studies on GPU kernel fusion for Transformers [11], [35], and it highlights how scheduling for Transformer matmuls becomes more complex when targeting different styles of custom hardware designs, including the Gemmini accelerator.

## IV. NEURAL ARCHITECTURE OPTIMIZATION

Another important avenue in full stack optimization of DNNs is optimizing DNN architectures and tailoring them for specific hardware platforms. However, the exponential search space of DNN architectures often makes it challenging to find an optimal architecture, even without considering the underlying hardware. To address this issue, automated neural architecture search (NAS) methods have been proposed to adapt DNNs for given hardware constraints. In this regard, we apply hardware-aware NAS to search for Transformer architectures that are optimal on the Gemmini-driven accelerator with better efficient and performance trade-offs. For a more detailed overview of hardware-aware NAS and its application to the Transformer architectures, please refer to Section 6 of our full paper [27].

*1) Experiment Setup:* As a baseline architecture, we use a 6-layer Transformer architecture with all other model configurations remaining the same as BERT-Base. We use language modeling on the WikiText-2 [31] as a training objective. To evaluate the model performance, we measured perplexity on the validation examples, where lower scores indicate better performance. The stand-alone baseline model was trained for 50 epochs with the Adam optimizer and a linear learning rate

scheduling with a peak learning rate of $\{5, 2, 1, 0.5\} \times 10^{-5}$. We use a sequence length of 512 and a batch size of 16.

For NAS, we adopt the BigNAS [47] strategy to train a supernet using the same training hyperparameters as the stand-alone training. The NAS search space is comprised of various combinations of the number of layers in $\{3, 4, 5, 6\}$, number of heads in $\{4, 6, 8, 10, 12\}$, hidden dimension in $[384, 768]$, and FFN dimension in $[768, 3072]$. Subsequently, we use evolutionary search for 40 iterations with a population size of 40 and mutation probability of 0.2 to search optimal subnets out of the fully trained supernet. After every iteration, only the subnets that are Pareto-optimal in EDP (energy-delay-product) and perplexity are retained. To measure the hardware cost, we use a lookup table-based method for quickly assessing the latency and energy consumption of each subnet on the target hardware, instead of using time-consuming RTL simulation. The lookup table contains Timeloop [33] simulated latency and energy numbers for each operation, which are then summed up to estimate the end-to-end values for the entire subnets. After the evolutionary search, the Pareto-optimal subnets are then evaluated with an RTL simulator to obtain a more precise estimation of the latency. For the energy measure, we continue to use the numbers from Timeloop. For the target hardware, we use Gemmini with the optimizations applied in Sec. II.

*2) Experiment Results:* We show the NAS Pareto-frontier results for EDP, latency and energy in Fig. 4 (blue curves) where each point corresponds to a different Transformer architecture found from the evolutionary search algorithm. Additionally, we plot the stand-alone trained baseline Transformer model trained as a reference (× mark). As can be seen in the EDP plot (Fig. 4 Left), the NAS framework allows us to obtain multiple Transformer architectures with better hardware cost to perplexity trade-offs. That is, it finds architectures with similar or even better perplexity, as compared to the baseline with smaller hardware costs.

Fig. 4 (Middle and Right) further illustrates latency and energy separately. As one can see, it is possible to attain a 1.4× reduction in latency versus the baseline Transformer with 0.1 point perplexity degradation. If one could tolerate 1 point degradation in perplexity, latency can be reduced by 2.4×. With regards to energy, one can attain a 1.6× improvement considering 0.1 point perplexity degradation, and 4.4× when allowing perplexity degradation of 1 point. Taking

both together, it is possible to reduce EDP by $2.2\times$ with just 0.1 point perplexity degradation, and $10.6\times$ with 1 point perplexity degradation. These examples illustrate the power of co-design in allowing practitioners to choose a combination that best matches their needs. It is important to note that this represents a single run of our co-design methodology on a specific hardware platform, and results may vary depending on the target hardware and optimization goals.

## V. CONCLUSION

While Transformer models have shown significant performance improvements, their growing size and run-time complexity present a critical challenge in efficient inference. In this work, we have demonstrated the benefits of a full stack approach by leveraging the advantages of co-design and co-optimization techniques across the stack. We adapted a CNN-oriented accelerator to efficient Transformer inference by supporting integer-only nonlinear operations [26] and re-balancing the memory hierarchy, which yielded a $39.6\times$ latency reduction. We also applied NAS to search for Pareto-optimal Transformer architectures given the tradeoff between EDP and perplexity, leading to a $10.6\times$ EDP reduction with minimal performance drop. Altogether, we have exhibited a $88.7\times$ latency improvement without a noticeable performance drop compared to a naive implementation without full-stack considerations. We have also demonstrated that unlike in CNNs, nonlinear operations in Transformers require careful consideration when performing operator fusion when targeting custom accelerators, e.g. systolic-array based architectures. We expect more improvement when we take this into consideration when designing the end-to-end full stack optimization pipeline. We refer interested readers to our full paper [27], which includes (1) a comprehensive analysis of Transformer workloads, (2) an extensive survey of the current hardware and software solutions on efficient Transformer inference, and (3) case studies to quantify the advantages of co-design and co-optimization techniques across the stack on full-stack Transformer inference.

## ACKNOWLEDGEMENTS

We acknowledge gracious support from Meta and in particular Michael Anderson, Satish Nadathur and Summer Deng, as well as Google Cloud, Google TRC team, and specifically Jonathan Caton, Prof. David Patterson, and Jing Li. Prof. Keutzer's lab is sponsored by Intel corporation, Intel VLAB team, Intel One-API center of excellence, as well as funding through BDD and BAIR. Sehoon Kim would like to acknowledge the support from Korea Foundation for Advanced Studies (KFAS). Amir Gholami was supported through funding from Samsung SAIT. Michael W. Mahoney would also like to acknowledge a J. P. Morgan Chase Faculty Research Award as well as the DOE, NSF, and ONR. Our conclusions do not necessarily reflect the position or the policy of our sponsors, and no official endorsement should be inferred.

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
