# OpenReview forum: "Full Stack Optimization of Transformer Inference"
_iscaconf.org/ISCA/2023/Workshop/ASSYST — ASSYST Oral_

### Official Review · Reviewer_3HkC · 2023-05-03
**Effective optimization of Transformer Inference by adopting and tuning existing optimization methods**

**Rating:** 6
**Confidence:** 3

**Review:**

The paper proposes a combination of hardware optimization and Neural architecture search to improve the speed-up of transformer architecture. Proposed method is a combination of I-Bert, layer fusing and NAS to optimize the inference of transformer. Each component contributes to the overall speed-up and efficiency of the inference.

**Review (Strengths/Weaknesses):**

Strengths:

++ The paper outlines the challenges correctly to motivate the solution. Intuitively, existing CNN accelerators are not optimized for transformer models. The paper tunes the gemini to support integer only I-Bert.

++ Fusion optimized scheduling is effective with significant improvement with MHA latency.

++ The paper is well-written.

Weaknesses:

-- The contribution from software DNN optimization is limited as NAS is a general and matured technique for hardware overhead optimization of DNN. Overall the novelty is limited.

-- evaluation section can be improved by providing more analysis specially the impact of model accuracy is not clear from the current evaluation plot in figure-4.

**Reviewer Expertise:**

Knowledgeable: I used to work in this area and/or I try to keep up with the literature but might not know the latest developments.

---

### Official Review · Reviewer_1YNx · 2023-05-03
**A good paper with many valuable finding. Backed up with reference and more experiment results will be appreciated.**

**Rating:** 4
**Confidence:** 5

**Review:**

**Summary**

The authors propose a full-stack optimization for transformer inference, which yields 88.7x speed-up.

**Review (Strengths/Weaknesses):**

**Strengths**
* I agree with all the finding that the authors had in performance bottleneck, quantization, HW-SW co-design, and so on. All the findings are valuable for the research community.

* I love that the authors are aware of the importance of full-stack optimization and are dedicating themselves to it.

**Weaknesses**
* Even though I could see many good takeaway in each sections, I find it hard to have a good grasp of what is the contribution of this paper  after reading. Did the authors propose something new? Is the contribution about sticking all these different techniques together and present the end-to-end result? Without clearly positioning the contribution, even though I like the takeaways of the paper, it is hard to for me to evaluate the work this paper have done.
* The related work section is missing. This is important.
* There is no comparisons to (STOA) related works in the experiments. It is important for the reader to set the ground on how this work improves over the baseline.

**Reviewer Expertise:**

Expert: I have written one or more papers on this topic and/or I currently work in this area.

---

### Official Review · Reviewer_CCjU · 2023-05-14
**Solid and timely work**

**Rating:** 7
**Confidence:** 5

**Review:**

### Summary:

The paper discusses the challenges in deploying Transformer models in latency-sensitive applications due to the increasing compute and bandwidth requirements. The authors propose a full-stack approach to optimize Transformer inference, considering hardware implications, nonlinear and linear operations, and neural architecture search. They adapt a CNN-oriented accelerator, introduce optimizations in memory configuration, and implement integer-only variants of nonlinear operations. The authors demonstrate significant latency reduction (up to 88.7×) without noticeable performance degradation compared to a naive implementation.

### Review:

The paper addresses an important issue in the field of neural network deployment, namely the increasing compute and memory bandwidth requirements of Transformer models. The authors present a comprehensive approach to optimizing Transformer inference, considering various aspects such as hardware architecture, nonlinear operations, linear operations, and neural architecture search. Using a full-stack optimization strategy is commendable, as it allows for maximizing performance gains across different layers of the system.

The adaptation of the Gemmini CNN accelerator for Transformer inference is a key contribution of this work. By modifying the accelerator and its software stack, the authors were able to improve the performance of Transformer models. The changes made to the memory configuration, particularly the adjustment of input/weight scratchpad and partial sum accumulator sizes, significantly reduced matmul latency.

The experiments on the integer-only BERT variant (I-BERT) is another notable contribution. By replacing floating-point nonlinear operations with integer polynomial approximations, the authors achieved faster and more efficient implementations in specialized accelerators. This approach eliminates the need for quantization and dequantization, allowing for fusion of operations and reducing overhead. The performance improvement of 39.6× achieved with I-BERT demonstrates the effectiveness of this optimization.

The application of neural architecture search (NAS) to find efficient and high-performance Transformer architectures is an exciting addition to the paper. By leveraging NAS, the authors discovered architectures that improved energy-delay product (EDP) by 10.6× with minimal performance drop. This demonstrates the potential of automated search methods in optimizing neural network architectures for specific hardware platforms.

Overall, the paper is well-structured and provides detailed insights into the optimization techniques employed to improve Transformer inference. The experimental results and performance comparisons are comprehensive, demonstrating the effectiveness of the proposed optimizations. The findings are significant and contribute to the ongoing efforts to make Transformer models more efficient for deployment in latency-sensitive applications.

**Review (Strengths/Weaknesses):**

### Suggestions for Improvements:

One of the paper's key contributions is the adaptation of a CNN-oriented accelerator to efficiently handle Transformer models. The authors introduce integer-only nonlinear operations and rebalance the memory hierarchy, resulting in a remarkable 39.6× reduction in latency. This modification demonstrates the benefits of tailoring hardware designs to the specific requirements of Transformers. However, the paper lacks a detailed analysis of the trade-offs introduced by these changes. It would be valuable to explore the impact of these modifications on other aspects, such as power consumption, scalability to larger models, and compatibility with existing hardware ecosystems.

Another critical aspect the paper addresses is the scheduling optimization for fusing high-arithmetic-intensity matmuls with low-arithmetic-intensity normalization operations. The authors propose fusion-optimized scheduling to overlap the execution of normalization operations with their preceding matmuls. While this strategy shows promising results for specific operations, such as query × key matmuls with Softmax, it worsens total latency by 27% for FFN W2 projection and LayerNorm overlapping. This finding highlights the complexity of scheduling for Transformer matmuls and emphasizes the need for further investigation into scheduling strategies that can achieve optimal performance across different hardware designs.

The paper also explores the application of hardware-aware NAS to search for optimal Transformer architectures on the modified accelerator. The NAS framework successfully discovers architectures with better hardware cost to perplexity trade-offs, enabling latency and energy consumption reductions. However, the evaluation of the NAS approach could be further strengthened. For example, the experimental setup lacks a comparison with other manually designed architectures. Furthermore, the paper only presents results for a specific hardware platform, and it remains unclear how the performance of the identified architectures would vary on different target hardware or optimization goals.

**Reviewer Expertise:**

Expert: I have written one or more papers on this topic and/or I currently work in this area.